# Enhancing adversarial robustness in Natural Language Inference using explanations

**Alexandros Koulakos, Maria Lymperaiou, Giorgos Filandrianos, Giorgos Stamou**
Artificial Intelligence and Learning Systems Laboratory
School of Electrical and Computer Engineering
National Technical University of Athens
koulakosalexandros@gmail.com, {marialymp, geofila}@islab.ntua.gr, gstam@cs.ntua.gr

## Abstract

The surge of state-of-the-art transformer-based models has undoubtedly pushed the limits of NLP model performance, excelling in a variety of tasks. We cast the spotlight on the underexplored task of Natural Language Inference (NLI), since models trained on popular well-suited datasets are susceptible to adversarial attacks, allowing subtle input interventions to mislead the model. In this work, we validate the usage of natural language explanation as a model-agnostic defence strategy through extensive experimentation: only by fine-tuning a classifier on the explanation rather than premise-hypothesis inputs, robustness under various adversarial attacks is achieved in comparison to explanation-free baselines. Moreover, since there is no standard strategy for testing the semantic validity of the generated explanations, we research the correlation of widely used language generation metrics with human perception, in order for them to serve as a proxy towards robust NLI models. Our approach is resource-efficient and reproducible without significant computational limitations.[1]

## 1 Introduction

Natural Language Inference (NLI) is a fundamental NLP task, aiming to define whether a hypothesis is entailed by, contradicts or remains neutral with respect to a given premise (Bowman et al., 2015). Despite primarily being a classification task, the subtle intricacies related to the semantic relationship of premise-hypothesis inputs with respect to the final label pose inherent challenges even for humans (Gururangan et al., 2018; Kalouli et al., 2021), causing annotation difficulties and thus data scarcity. Within the rapidly evolving NLP landscape, there are still several unsolved challenges, especially concerning the usage of Large Language

Models (LLMs) for NLI, which are yet unable to fully capture the semantic sophistications of the task (Gubelmann et al., 2023; Kavumba et al., 2023).

At the same time, explainability remains a point of reference for state-of-the-art (SoTA) NLP (Danilevsky et al., 2021; Liao and Vaughan, 2023); however, it holds an even more crucial position for NLI, as stated in the seminal work of Camburu et al. (2018), where authors hint that generating an intermediate explanation before predicting the final label is adequate for robustness enhancement. This is a fundamental claim, as NLI models are widely susceptible to adversarial attacks (Alzantot et al., 2018; Zhang et al., 2019b; Jin et al., 2020). Yet, to the best of our knowledge, there is no prior work attempting to solely harness explanations for adversarial defence, in order to answer whether this claim holds or not. The additional power of intermediate explanations is that they shed some light on the black-box nature of NLI models, providing information regarding the semantic relationship between the premise and the hypothesis. Under this breakdown of the NLI process, the weight is shifted towards producing a semantically valuable and correct explanation, which is highly associated with the final label, as we will demonstrate later in this paper. Therefore, without exploiting any other mechanism rather than the intermediate explanations, we are able to open the black-box while simultaneously rendering it more robust.

Overall, in this work, we propose a very simple, yet effective approach to tackle adversarial brittleness of NLI: we leverage the ExplainThenPredict framework proposed in Camburu et al. (2018) to acquire explanations derived from given premise-hypothesis input pairs, based on which we predict the final label. To further promote the simplicity of our method, we only exploit smaller language models for explanation generation, as well as for classification in the entailment/neutral/contradiction

---

classes, proving that despite not being the most powerful learners, they are adequate in proving the robustness-enhancing power of explanations. Specifically, our contributions are:

- We experimentally prove that generating explanations leads to more robust NLI classification under various adversarial attacks.

- In order to facilitate (approximate) explanation evaluation, we provide an association between metrics for linguistic quality of explanations and model robustness, as verified by humans.

## 2 Related work

**Natural Language Inference (NLI)** is a core NLP task tied to language understanding, although it remains comparatively underexplored. The breakthrough introduced with the SNLI (Stanford Natural Language Inference) dataset (Bowman et al., 2015) inspired several approaches over the years, ranging from LSTM-based (Chen et al., 2017) to transformer-based ones (Devlin et al., 2019; Zhang et al., 2019c; Sun et al., 2020; Radford and Narasimhan, 2018; He et al., 2023). Incorporating explanations in the e-SNLI variant (Camburu et al., 2018) introduced a favourable research line focused on interpretable NLI (Chen et al., 2021; Stacey et al., 2021, 2022; Yu et al., 2022; Yang et al., 2023a; Abzianidze, 2023; Yang et al., 2023b), with faithfulness of explanations (Kumar and Talukdar, 2020; Zhao and Vydiswaran, 2020; Lyu et al., 2022; Sia et al., 2023) serving as a core research endeavour, tied to present NLI limitations.

**Adversarial Robustness** is a major concern in NLP (Goyal et al., 2023; Goel et al., 2021) calling for successful detection and creation of defence strategies (Shen et al., 2023; Sabir et al., 2023; Yang and Li, 2023). Crafting adversarial attacks (Jin et al., 2020; Li et al., 2020; Liu et al., 2022; Asl et al., 2024) reveals weak spots of models in cases they return unreasonably deviating outputs with respect to the semantic minimality of input perturbations. In general, the quest for robustness may require some sacrifice in accuracy (Tsipras et al., 2018; Zhang et al., 2019a), at least under certain scenarios that cannot be satisfied by theoretical guarantees (Yang et al., 2020; Pang et al., 2022; Chowdhury and Urner, 2022; Chen et al., 2024), such as in black-box settings where adequate engineering regarding training details and

hyperparameters is not feasible. This trade-off has not been extensively studied in NLP or at least in various black-box cases, therefore it is unknown if it holds when studying them in conjunction.

Despite the suggested incorporation of explanations for robust NLI (Camburu et al., 2018), this topic has not received much attention yet, with only a few notable exceptions (Alzantot et al., 2018; Nakamura et al., 2023), while the utilized explanations benefit other robustness-related research questions, such as the robustness of in-context learning in LLMs (He et al., 2023). We highly acknowledge this research gap, promoting the exploitation of explanations as a model-agnostic strategy to enhance NLI robustness under adversarial attacks.

## 3 On the use of intermediate explanations

In the core of our approach lies the ExplainThenPredict framework (Camburu et al., 2018) that instead of predicting the final entailment (E)/neutral (N)/contradiction (C) label using the input premise and hypothesis, it generates an intermediate explanation in natural language, which serves as an input to a classifier to decide the final label.

As a first step, a Seq2Seq model receives the premise **P** and the hypothesis **H** and outputs a free-form explanation **e**, which aims to justify the semantic relationship between them under an entailment/neutral/contradiction format. For example, given **P**: "A Land Rover is being driven across a river" and **H**: "A vehicle is crossing a river", the Seq2Seq stage generates an explanation **e**: "Land Rover is a vehicle". In the second step, an Expl2Label classifier defines the output label $L \in \{E, N, C\}$, leveraging the "hints" provided in the explanation. In the aforementioned example, given **e**: "Land Rover is a vehicle", the Expl2Label classifier outputs *Entailment* as the final label.

Notably, Seq2Seq and Expl2Label are fine-tuned independently and are only joined during inference, acting as a black-box model overall.

### 3.1 Experimental setup

Our experimentation is applied on the e-SNLI dataset (Camburu et al., 2018). We focus on testing affordable models due to the computational burden imposed by fine-tuning the Seq2Seq and Expl2Label models on the derived explanations, aiming to provide a lightweight solution that is reproducible regardless of hardware limitations. More specifically, for the Seq2Seq stage we utilize

| | BERT2GPT | ALBERT2GPT | DISTILBERT2GPT | ROBERTA2GPT |
|---|---|---|---|---|
| fine-tuning time ($\downarrow$) | 12 hrs & 20 mins | 12 hrs & 16 mins | **9 hrs & 35 mins** | 12 hrs & 29 mins |
| meteor ($\uparrow$) | 0.5332 | **0.5591** | 0.5393 | 0.5509 |
| bert-score ($\uparrow$) | 0.8707 | 0.8742 | 0.8701 | **0.8744** |
| rouge ($\uparrow$) | 0.5885 | 0.6005 | 0.5859 | **0.6011** |
| bleu ($\uparrow$) | 0.3859 | 0.3911 | 0.3719 | **0.3992** |
| % correct explanations ($\uparrow$) | 76.14% | 73.33% | 72.53% | **77.17%** |

Table 1: Seq2Seq scores during inference using the various encoders and GPT2 decoder. The optimal values and fine-tuning time needed among all 4 Seq2Seq models are denoted with **bold** font.

encoder-decoder structures, with GPT-2[2] (Radford et al., 2019) serving as the decoder, while BERT[3] (Devlin et al., 2019), ALBERT[4] (Lan et al., 2020), DistilBERT[5] (Sanh et al., 2019) and RoBERTa[6] (Liu et al., 2019) are placed as encoders, one at a time. Regarding the Expl2Label stage, we exploit a single, yet effective BERT model with a classification head which achieves high classification accuracy in NLI labels.

For explanation evaluation, we utilize text generation metrics, including METEOR (Banerjee and Lavie, 2005), ROUGE (Lin, 2004), BLEU (Papineni et al., 2002) and BERTScore (Zhang et al., 2020) using the 3 provided ground-truth e-SNLI explanations as references. Nevertheless, these metrics do not directly reflect the explanation quality in terms of semantic faithfulness. For this reason, we manually evaluate[7] the semantic faithfulness of explanations and measure the correlation of text generation metrics with our annotations, finally recommending the most suitable metric as a proxy for human-interpretable explanation quality (App. A). The final NLI label is evaluated based on accuracy.

All experiments are conducted using a single NVIDIA Volta V100 GPU. The batch size is set to 32, the encoder max length is selected to be 128 tokens for any encoder, while the decoder max length is 64 tokens for the GPT-2 decoder. Fine-tuning is performed for 5 epochs, while greedy decoding is the default text generation strategy.

[2]https://huggingface.co/openai-community/gpt2 [GPT2-small (124M)]
[3]https://huggingface.co/google-bert/bert-base-uncased [bert-base-uncased (110M)]
[4]https://huggingface.co/albert/albert-base-v2 [albert-base-v2 (12M)]
[5]https://huggingface.co/distilbert/distilbert-base-uncased [distilbert-base-uncased (66M)]
[6]https://huggingface.co/FacebookAI/roberta-base [roberta-base (125M)]
[7]Conducted by the authors on 100 samples to ensure the validity of results, due to the inherent difficulty of associating semantic relatedness of explanation with input **P** & **H**.

We regard two NLI classification baselines: first, the explanation-free setup of directly predicting the output label by feeding **P-H** pairs in a BERT-based classifier. Second, we compare with training BERT with ground-truth explanations from the e-SNLI dataset using the same hyperparameters and hardware mentioned above.

### 3.2 Explanation generation results

As a first step, we evaluate the quality of the Seq2Seq stage using the available combinations of encoders with the GPT2 decoder, namely BERT2GPT, ALBERT2GPT, DISTILBERT2GPT and ROBERTA2GPT. We report the aforementioned text generation metrics, as well as the explanation accuracy, which is defined as the percentage of explanations that semantically represent the ground-truth label according to our manual annotation. Related results are presented in Table 1. We also report time needed for fine-tuning.

Overall, we can easily observe that ROBERTA2GPT scores higher in terms of most text generation metrics, as well as the number of semantically correct explanations. The time needed for fine-tuning is ∼12 hours in most cases, with DISTILBERT2GPT serving as a more efficient alternative due to its distillation process, with slightly lower text generation quality and ∼5% sacrifice in explanation accuracy.

### 3.3 NLI classification results

Given the explanations produced in the previous step as inputs, the Expl2Label module decides upon the final E/N/C label. Regarding baselines, we first fine-tune an explanation-free BERT model using input **P-H** pairs. Consequently, we fine-tune BERT on the ground-truth e-SNLI explanations. Related baseline results are reported in Table 2.

Focusing on the 2nd row, BERT fine-tuned on ground-truth explanations achieves an accuracy score of 97.47%; this significantly high accuracy

| Baseline | Fine-tuning time | Accuracy |
|---|---|---|
| Explanation-free BERT | ~ 6 hrs | 90.13% |
| Ground-truth BERT **e** | 5 hrs & 42 mins | 97.47% |

Table 2: Fine-tuning time and accuracy of baselines.

denotes the strong association between the explanation and the final label, even *without any* information regarding the corresponding **P** and **H**. Even though this claim further supports the ExpainThenPredict decomposition, there are some shortcomings related to the format of the explanation, so that sometimes the same explanation justifies diverging ground-truth labels stemming from different hypotheses, as demonstrated in Table 3.

| P-H pair | L | e |
|---|---|---|
| **P**: A woman is in the park
**H**: A person is in the park | E | A woman is a person |
| **P**: A woman is in the park
**H**: There is no person in the park | C | A woman is a person |

Table 3: The same explanation can justify a different label depending on the input **P** and **H**. For the contradiction pair, one could also explain that "There can be no person in the park if a woman is in the park" which is more indicative of contradiction (Camburu et al., 2018).

By accepting such imperfections, and recognizing that formulating an informative and correct explanation is a separate research problem, we proceed by fine-tuning the same BERT architecture using the ground truth explanations **e** from the e-SNLI dataset, while for inference, we use the explanations derived from the previously described Seq2Seq variants. In Table 4, we report accuracy scores (overall & per label) for each Seq2Seq explanation followed by the fine-tuned BERT.

| | BERT
2GPT | ALBERT
2GPT | DISTILBERT
2GPT | ROBERTA
2GPT |
|---|---|---|---|---|
| acc | 86.72% | 85.45% | 85.15% | **87.97%** |
| acc (E) | 89.13% | 86.76% | 87.29% | **90.17%** |
| acc (C) | 90.42% | 88.35% | 85.82% | **91.69%** |
| acc (N) | 80.4% | 81.14% | **82.20%** | 82.01% |

Table 4: ExplainThenPredict inference results using BERT as the Expl2Label classifier. Best results among all 4 ExplainThenPredict variants are denoted in **bold**.

It becomes evident that the generated explanations result in a decrease of overall accuracy scores (1st row of Table 4) in comparison to the baselines (Table 2). However, in the next section we will

highlight the real value of such a sacrifice.

## 4 Adversarial Attacks

We stress the robustness of the ExplainThenPredict pipeline by performing targeted adversarial attacks either on **P** or **H** independently. The outline of our proposed approach is illustrated in Figure 1.

We focus on applying minimal interventions that influence the semantics of the inputs, resulting in adversarial **P**→**P\*** or **H**→**H\*** transitions. Such interventions consequently lead to **e**→**e\*** transitions, which finally affect the outcome of the BERT classifier, resulting in a **L**→**L\*** transition of the final predicted label. Given the semantic minimality of the intervention, a **L**→**L\*** change denotes a possible excessive attachment on the words of **P** or **H** rather than their meaning, indicating a vulnerable behaviour in terms of classification robustness.

The intervention needs to be targeted, since altering the predicted NLI label is significant to view an attack as "successful": a negligible semantic intervention erroneously leads to a change of the NLI prediction; in the ExplainThenPredict case this change happens because the attack on **P/H** affected the intermediate explanation **e** (if no change had occurred on the **e**, no outcome change could be possible). Therefore, we materialize the desired attacks using attack recipes from BERT-Attack (Li et al., 2020) and TextFooler (Jin et al., 2020), which serve as SoTA word-level editors, providing the requested determinism that guarantees the minimality of edits, while preserving the meaning and syntax of the attacked sentence. By attacking the explanation-free baseline, as well as the ExplainThenPredict variants we are able to measure the *attack success rate*, i.e. the ratio of attempted attacks that successfully produce adversarial examples in each case. Therefore, the higher the *attack success rate*, the more vulnerable the model is against such interventions. Moreover, we calculate the *after-attack accuracy*, corresponding to the percentage of inputs that were unsuccessfully attacked and correctly classified, with higher values denoting more robust models. The *attack success rate* and *after-attack accuracy* accuracy metrics hold an inverse relationships, with more robust models presenting lower *attack success rate* and higher *after-attack accuracy*. For the sake of completeness, we further report the *average number of queries*, which denotes the attack efficiency, corresponding to the number of attacks that the attacker needs to

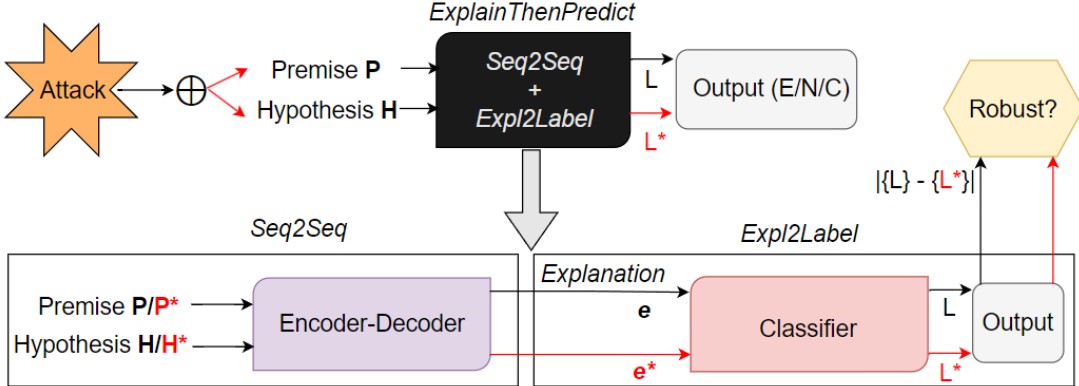

Figure 1: Outline of our approach: we enforce an adversarial perturbation on either **P** or **H** of ExplainThenPredict.

perform in order to change the outcome. Higher values indicate that the attacker needs to place more effort in order to achieve the alternative outcome, implying comparatively advanced resistance from the side of the attacked model. This experimentation aims to conclude under which circumstances the ExplainThenPredict framework leads to more robust NLI classification and how we can obtain some guarantees regarding advanced robustness.

### 4.1 Experimental setup

Regarding TextFooler, we attempt balancing diversity of interventions and maintaining similarity with the original input. To this end, we focus on the diversity-related hyperparameter $N$ that refers to the number of candidates needed to substitute a vulnerable word; $N$ is controlled using the max_candidates hyperparameter in TextFooler documentation, which is set to 50 according to Jin et al. (2020). In the meanwhile, the similarity hyperparameter $\delta$ that dictates the degree of semantic closeness between the intervened text and the original one sets the minimum threshold for an intervention to form a valid adversarial in terms of semantic minimality. We set the corresponding TextFooler documentation hyperparameter max_candidates to 0.7 (recommended from Jin et al. (2020)) and 0.75, examining balancing the diversity-similarity trade-off in the first case, while also exploring favouring similarity over diversity in the second case.

As for BERT-Attack, the hyperparameter $K$ defines the number of candidates needed to substitute a vulnerable word, equivalent to TextFooler's $N$ hyperparameter, with higher $K$ values imposing more diverging synonym substitutions, thus expecting to increase the *attack success rate*. To explore the influence of this variability, we experiment with

recommended values of $K \in \{6, 8\}$.

We remain within the black-box setting, since the attacks are enforced on the input **P/H**, while we probe its influence on the **L**→**L\*** change.

### 4.2 Results

We report results regarding TextFooler attacks on **P** or **H** at a time in Table 5. It is easily noticeable that our original claim holds: under TextFooler attacks, the *attack success rate* of ExplainThenPredict is lower in comparison to the explanation-free baseline, implying advanced robustness when explanations are incorporate within the pipeline. Moreover, Figure 2 reports the % decrease on *attack success rate* for all ExplainThenPredict variants.

Similarly, results for the BERT-Attack recipe are presented in Table 6 and Figure 3, verifying the patterns of increased robustness when generated explanations are utilized, as in the TextFooler case.

Regarding TextFooler attacks, ROBERTA2GPT consistently arises as the most robust Seq2Seq module for explanation generation, scoring higher in *after-attack accuracy* and *average number of queries* needed, while presenting lower scores in *attack success rate*. By also comparing the ROBERTA2GPT results with the metrics related to the other Seq2Seq models, we can conclude that **the choice of Seq2Seq model matters**, since an insufficient explanation generator may lead to decreased ExplainThenPredict robustness even in comparison with the explanation-free baseline.

The robustness guarantees are strongly associated with the quality of the explanations themselves: the linguistic quality of explanations, as well as the human perception of correctness (Table 1) are consistently correlated with ExplainThen-Predict model robustness, with RoBERTa arising

| | | Baseline | BERT2GPT | ALBERT2GPT | ROBERTA2GPT | DISTILBERT2GPT |
|---|---|---|---|---|---|---|
| | Original accuracy (↑) | **90.13**% | 86.72% | 85.45% | 87.97% | 85.15% |
| | **TextFooler (target sentence: P)** | | | | | |
| $N = 50, \delta = 0.7$ | After-attack accuracy (↑) | 24.93% | 27.76% | 24.2% | **28.74**% | 24.08% |
| | Attack success rate (↓) | 72.16% | 67.99% | 71.69% | **67.33**% | 71.1% |
| | Avg num queries (↑) | 43.74 | 44.1 | 41.75 | **44.57** | 42.16 |
| | **TextFooler (target sentence: H)** | | | | | |
| | After-attack accuracy (↑) | 10.33% | 13.86% | 12.68% | **16.31**% | 11.83% |
| | Attack success rate (↓) | 88.46% | 84.01% | 85.16% | **81.46**% | 86.11% |
| | Avg num queries (↑) | 24.3 | 24.6 | 25.05 | **25.92** | 24.16 |
| | **TextFooler (target sentence: P)** | | | | | |
| $N = 50, \delta = 0.75$ | After-attack accuracy (↑) | 33.22% | 35.2% | 30.92% | **36.18**% | 31.1% |
| | Attack success rate (↓) | 62.9% | 59.41% | 61.5% | **58.88**% | 61.2% |
| | Avg num queries (↑) | 37.02 | 36.68 | 35.11 | **37.14** | 35.49 |
| | **TextFooler (target sentence: H)** | | | | | |
| | After-attack accuracy (↑) | 15.89% | 18.6% | 18.19% | **21.85**% | 16.73% |
| | Attack success rate (↓) | 82.26% | 78.55% | 78.71% | **75.16**% | 80.35% |
| | Avg num queries (↑) | 20.64 | 20.64 | 21.08 | **21.67** | 20.27 |

Table 5: Attack results synopsis for attacking **P** or **H** using TextFooler. **Bold** values denote best results (row-wise).

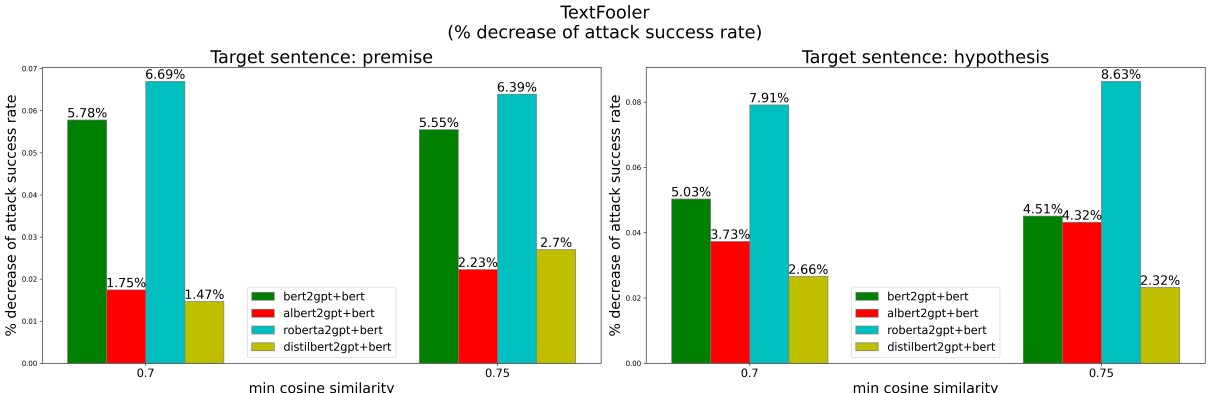

Figure 2: Visualization of the % attack success rate decrease achieved by the ExplainThenPredict model variations under **TextFooler** attack setting.

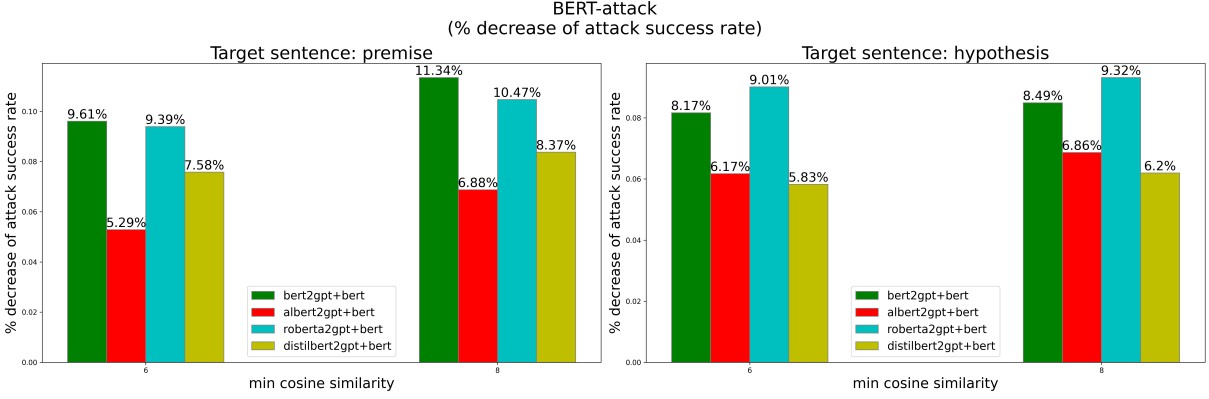

Figure 3: Visualization of the % attack success rate decrease achieved by the ExplainThenPredict model variations under **BERT-attack** attack setting.

|  |  | Baseline | BERT2GPT | ALBERT2GPT | ROBERTA2GPT | DISTILBERT2GPT |
|---|---|---|---|---|---|---|
|  | Original accuracy (↑) | **90.13**% | 86.72% | 85.45% | 87.97% | 85.15% |
|  | **BERT-Attack (target sentence: P)** | | | | | |
| $K = 6$ | After-attack accuracy (↑) | 19.26% | 25.18% | 21.92% | **25.4**% | 23.37% |
|  | Attack success rate (↓) | 78.5% | **70.96**% | 74.35% | 71.13% | 72.55% |
|  | Avg num queries (↑) | 26.96 | 29.77 | 28.52 | **29.93** | 29.19 |
|  | **BERT-Attack (target sentence: H)** | | | | | |
|  | After-attack accuracy (↑) | 9.16% | 15.23% | 13.48% | **16.11**% | 13.16% |
|  | Attack success rate (↓) | 89.77% | 82.44% | 84.23% | **81.68**% | 84.54% |
|  | Avg num queries (↑) | 15.28 | 16.24 | 16.3 | **16.59** | 16.03 |
|  | **BERT-Attack (target sentence: P)** | | | | | |
| $K = 8$ | After-attack accuracy (↑) | 14.79% | **22.54**% | 19.02% | 22.22% | 20.02% |
|  | Attack success rate (↓) | 83.48% | **74.01**% | 77.74% | 74.74% | 76.49% |
|  | Avg num queries (↑) | 31.27 | **35.9** | 33.84 | 35.68 | 34.88 |
|  | **BERT-Attack (target sentence: H)** | | | | | |
|  | After-attack accuracy (↑) | 4.87% | 11.68% | 10.19% | **12.53**% | 9.61% |
|  | Attack success rate (↓) | 94.57% | 86.54% | 88.08% | **85.76**% | 88.71% |
|  | Avg num queries (↑) | 17.37 | 18.93 | 19.03 | **19.46** | 18.69 |

Table 6: Attack results synopsis for attacking **P** or **H** using BERT-Attack. **Bold** values denote best results (row-wise).

as the most potent encoder[8], as all other components of ExplainThenPredict remain invariant. Thus, since the choice of Seq2Seq module matters, we safely conclude that **optimizing explanation quality results in advanced ExplainThenPredict robustness**. As a byproduct of this observation, we can state that leveraging ExplainThenPredict to advance NLI robustness is not sufficient on its own, and the weight needs to be shifted towards producing more faithful and linguistically correct explanations.

Some interesting patterns occur from the analysis of BERT-Attack results in Table 6: in this case, all reported metrics associated with employing ExplainThenPredict are better in comparison to the explanation-free baseline. This behavior denotes that even lower-quality intermediate explanations are sufficient for boosting NLI robustness, and the evaluation of explanation quality and faithfulness is not necessary for guaranteeing robustness.

The attacks are consistently more effective when targeting **H** rather than **P** regardless the attacker utilized each time, or its hyperparameters. We delve into qualitative examples to understand this pattern.

### 4.3 Qualitative Analysis

We present some examples regarding how an attack from TextFooler (Table 9 in App. B) and BERT-

Attack (Table 10 in App. B) influences the input **P** and **H** at a time, altering the intermediate **e** and the final label **L**.

In most cases, the form the explanation receives based on the input **P** and **H** significantly defines the final label: the entailment explanation format of "X is Y" or tautological statements such as "if X is Y, then X is Y" are highly associated with E label. On the other hand, explanation formats such as "X is not Y" conclude towards C label. Finally, statements like "X is not necessarily Y" or similar lead to N label. Notably, the explanation simplifies the NLI classification task by connecting the semantic meaning between **P** and **H**, acting as an intermediate reasoning step that enhances clarity in a concise manner. To this end, we can easily observe how input modifications influence this format of the explanations, which ultimately drives the selection of the label **L\***. Even synonym substitutions on behalf of the attacker easily derail the semantic connection between **P** and **H**, which is reflected on the generated explanation **e\***.

Regarding the advanced sensitivity observed on model robustness when **H** is attacked (Tables 5, 6), we assume that this is due to the shorter length of **H**; therefore, intervened semantics of **H** cannot be matched with their counterparts present in **P**. On the other hand, if **P** is attacked, there is a possibility that intervened semantics are not part of **H** at all, therefore the initial reasoning path remains valid.

In most cases, the generated explanations pro-

---

[8]An interesting future work would be to experiment with baseline classifier architectures other than BERT (e.g. ALBERT, DistilBERT, RoBERTa) and examine if we get similar results.

|  | Premise P | Hypothesis H | Label L | Explanation e |
|---|---|---|---|---|
| Explain ThenPredict | A young family enjoys feeling ocean waves lap at their feet | A family is at the beach | E | A young family is a family. Ocean waves are at the beach. |
|  | **Premise P** | **Hypothesis H** | **Label L** | **Explanation e** |
| Expl-free | A young family enjoys feeling ocean waves lap at their feet | A family is at the beach | E | N/A |
|  | **Premise P\*** | **Hypothesis H** | **Label L\*** | **Explanation e** |
|  | A young familia enjoys feeling ocean waves lap at their feet | A family is at the beach | N | N/A |
| Explain ThenPredict | **Premise P** | **Hypothesis H** | **Label L** | **Explanation e** |
|  | A couple walks hand in hand down a street | A couple is walking together | E | If they are walking hand in hand, they are walking together. |
| Expl-free | **Premise P** | **Hypothesis H** | **Label L** | **Explanation e** |
|  | A couple walks hand in hand down a street | A couple is walking together | E | N/A |
|  | **Premise P\*** | **Hypothesis H** | **Label L\*** | **Explanation e** |
|  | A pair walks hand in hand down a street | A couple is walking together | N | N/A |

Table 7: Example instances where the explanation-based models manage to resist the attack compared to the explanations-free baseline. Red color denotes the words attacked.

vide meaningful information regarding the **P-H** semantic relationship, even though they may sometimes be redundant. Nevertheless, in an ideal, fully robust setting, the explanation format, which betrays the final label, should not be altered after semantically minimum interventions. Despite the reported instabilities of Tables 9, 10 in App. B, many instances correctly retain their label after attack in comparison to the explanation-free baseline which remains way more brittle. This is clearly illustrated in Table 7, where we can see that even with identical premise and hypothesis pairs, the explanation-free baseline model is deceived, while the prediction of the explanation-based model remains unaffected, due to the accurate and high-quality generated explanation.

## 5    Conclusion

In this work, we delve into the underexplored field of NLI robustness. We experimentally prove that the robustness of NLI models against adversarial attacks can be boosted by solely generating intermediate explanations. Furthermore, we demonstrate that linguistic quality and human perception of faithfulness are strongly correlated with advanced robustness of the final model, drawing the attention to explanation evaluation as the natural next step in advancing trustworthy and interpretable NLI.

## Broader Impacts and Ethics

This work aims to advance the trustworthiness of NLI predictions providing interpretability and robustness by merely generating intermediate explanations before the final classification. We view our work as a starting point towards more capable, interpretable, efficient and reliable NLI models. The quality of the explanations is a crucial factor towards this goal, with possible concerns revolving around the degree of trust we should pose on possibly unfaithful explanations, even though quantitative results support their beneficial usage.

## Limitations

Our work serves as a primary investigation of the unexplored explanation-based NLI robustness under adversarial attack, proving there are many related research questions to be addressed. We believe that the most prominent limitation is to ensure faithfulness of explanations with respect to input premise and hypothesis, as well as the output label. To this end, searching, generating and evaluating faithful explanations (Gat et al., 2023; Sia et al., 2023) is the key to advance the performance and robustness of NLI models. A parallel concern lies on the annotation difficulty of NLI associations on its own (Kalouli et al., 2021), which somehow limits data abundance and therefore models and

evaluation methods, a fact that we verify through our manual annotation process for NLI explanations. As a secondary thought, experimentation using state-of-the-art LLMs may benefit the quality of explanations -at least from the linguistic viewpoint- even though there are no guarantees regarding their faithfulness; nevertheless, advancements in LLM reasoning (Qiao et al., 2023; Giadikiaroglou et al., 2024) may offer faithful explanations as a natural byproduct. On the other hand, exploiting LLMs requires high-end computational resources or paid schemes, thus significantly reducing accessibility.

## Acknowledgments

The research work was supported by the Hellenic Foundation for Research and Innovation (HFRI) under the 3rd Call for HFRI PhD Fellowships (Fellowship Number 5537).

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

# A Human annotation of explanations

Our manual annotation regards 100 samples from the e-SNLI dataset, containing the premise **P**, the hypothesis **H**, the explanation **e** and the ground-truth label **L**. We collect results from all combinations regarding the encoder of the Seq2Seq stage, and we evaluate whether the explanation is semantically correct in terms of the input **P** and **H**, as well as the ground-truth label. Some examples of the manual annotation are demonstrated in Table 8.

# B Qualitative results

Tables 9 and 10 present some qualitative examples from the ExplainThenPredict scenario that illustrate how an attack stemming from the TextFooler and BERT-Attack recipes influences the input components **P** and **H** at a time, thus altering the intermediate prediction **e** as well as the final label **L**.

| Premise P | Hypothesis H | Gold label | Generated explanation | Does the explanation fully justify the label? |
|---|---|---|---|---|
| A blond headed child in yellow boots and yellow jacket vest playing in the gravel with his pail, shovel and trucks | A blonde child is playing | E | A child playing is the same as a child playing | No |
| A blond headed child in yellow boots and yellow jacket vest playing in the gravel with his pail, shovel and trucks | A blonde child is playing | E | A blond headed child is a type of blond child and playing in the gravel is a type of playing | Yes |
| An elderly woman wearing a skirt is picking out vegetables at a local market | A young girl is blowing bubbles | C | An elderly woman is not a young girl | No |
| An elderly woman wearing a skirt is picking out vegetables at a local market | A young girl is blowing bubbles | C | An elderly woman is not a young girl. Picking out vegetables is not the same as blowing bubbles | Yes |

Table 8: Manual annotation examples from our sampled collection. In case of entailment, we consider an explanation accurate, if it includes all the reasons why the hypothesis is entailed by the premise. In case of contradiction, an explanation is accurate, if it includes all the reasons why the hypothesis contradicts the premise.

| | TextFooler (target sentence: P), $\delta$=0.75 | | | |
|---|---|---|---|---|
| | **Premise P** | **Hypothesis H** | **Explanation e** | **Label L** |
| Original | This church choir sings to the masses as they sing joyous songs from the book at a church | A choir singing at a baseball game | The church cannot be at a baseball game and at a church at the same time | C |
| | **Premise P\*** | **Hypothesis H** | **Explanation e\*** | **Label L\*** |
| Attacked | This clergy choir sings to the masses as they sing celebratory songs from the book at a clerical | A choir singing at a baseball game | The choir singing is not necessarily at a baseball game | N |
| | **Premise P** | **Hypothesis H** | **Explanation e** | **Label L** |
| Original | An old man with a package poses in front of an advertisement | A man poses in front of an ad | An advertisement is an ad | E |
| | **Premise P\*** | **Hypothesis H** | **Explanation e\*** | **Label L\*** |
| Attacked | An old fella with a package poses in front of an ad | A man poses in front of an ad | An old fella is not a man | C |
| | TextFooler (target sentence: H), $\delta$=0.75 | | | |
| | **Premise P** | **Hypothesis H** | **Explanation e** | **Label L** |
| Original | A woman with a green headscarf, blue shirt and a very big grin | The woman is young | Not all women are young | N |
| | **Premise P** | **Hypothesis H\*** | **Explanation e\*** | **Label L\*** |
| Attacked | A woman with a green headscarf, blue shirt and a very big grin | The woman is youthful | A woman with a green headscarf and a very big grin is youthful | E |
| | **Premise P** | **Hypothesis H** | **Explanation e** | **Label L** |
| Original | An old man with a package poses in front of an advertisement | A man walks by an ad | Poses and walks are not the same | C |
| | **Premise P** | **Hypothesis H\*** | **Explanation e\*** | **Label L\*** |
| Attacked | An old man with a package poses in front of an advertisement | A man strolls by an ad | Strolls is another way to say poses. An ad is an advertisement | E |

Table 9: Qualitative results for attacking **P/H** using TextFooler. Red color indicates changes induced by TextFooler.

| | **BERT-Attack (target sentence: P), $K$=8** | | | |
|---|---|---|---|---|
| | **Premise P** | **Hypothesis H** | **Explanation e** | **Label L** |
| Original | A young family enjoys feeling ocean waves lap at their feet | A family is at the beach | A family enjoys the ocean waves at the beach | E |
| | **Premise P\*** | **Hypothesis H** | **Explanation e\*** | **Label L\*** |
| Attacked | A young family enjoys feeling the waves lap at their feet | A family is at the beach | A family enjoying the waves at their feet is not necessarily at the beach | N |
| | **Premise P** | **Hypothesis H** | **Explanation e** | **Label L** |
| Original | Two children are laying on a rug with some wooden bricks laid out in a square between them | Two children are on a rug | If children are laying on a rug, then they are on a rug | E |
| | **Premise P\*** | **Hypothesis H** | **Explanation e\*** | **Label L\*** |
| Attacked | Two children are laying on a mat with some wooden bricks laid out in a square between them | Two children are on a rug | The children are either laying on a mat or on a rug | C |
| | **BERT-Attack (target sentence: H), $K$=8** | | | |
| | **Premise P** | **Hypothesis H** | **Explanation e** | **Label L** |
| Original | An old man with a package poses in front of an advertisement | A man walks by an ad | Poses and walks are not the same | C |
| | **Premise P** | **Hypothesis H\*** | **Explanation e\*** | **Label L\*** |
| Attacked | An old man with a package poses in front of an advertisement | A man steps by an ad | Poses in front of an advertisement is the same as steps by an ad | E |
| | **Premise P** | **Hypothesis H** | **Explanation e** | **Label L** |
| Original | One tan girl with a wool hat is running and leaning over an object, while another person in a wool hat is sitting on the ground | A man watches his daughter leap | The two people are not necessarily a man and the girl is not necessarily his daughter | N |
| | **Premise P** | **Hypothesis H\*** | **Explanation e\*** | **Label L\*** |
| Attacked | One tan girl with a wool hat is running and leaning over an object, while another person in a wool hat is sitting on the ground | A man sees his daughter leap | The two people are either a man and a woman, or a man and his daughter | C |

Table 10: Qualitative results for attacking **P/H** using BERT-Attack. Red color denotes words attacked by BERT-Attack.