# OpenReview forum: "Enhancing adversarial robustness in Natural Language Inference using explanations"
_EMNLP/2024/Workshop/BlackBoxNLP — BlackboxNLP 2024_

### Official Review · Reviewer_J88V · 2024-09-07

**Overall Assessment:** 3
**Confidence:** 2

**Best Paper:**

1

**Best Paper Justification:**

na

**Comments Questions Suggestions And Typos:**

na

**Paper Summary:**

The study focuses on the underexplored task of Natural Language Inference (NLI), highlighting how Transformer-based models, despite their strong performance, remain vulnerable to adversarial attacks. The research proposes using natural language explanations as a model-agnostic defense strategy.

**Summary Of Strengths:**

Paper is written well.
It shows that fine-tuning a classifier on explanations, rather than on premise-hypothesis inputs, improves robustness against adversarial attacks compared to models without explanations.

**Summary Of Weaknesses:**

Recent models should be test instead of GPT2

---

### Official Review · Reviewer_NrVf · 2024-09-15

**Overall Assessment:** 4
**Confidence:** 3

**Best Paper:**

1

**Best Paper Justification:**

N/A

**Comments Questions Suggestions And Typos:**

* The results table  4 and 5 arrangement is confusing; the bold numbers denote a row-wise high value instead of a column. It would be easier to follow if the tables are transposed.

**Paper Summary:**

The authors provide perspective on adversarial attacks on NLI systems and how a two-stage explain then predict paradigm might be a direction towards a defence.

**Summary Of Strengths:**

* This paper is thorough and analyzes the impact of adversarial attacks on-premise and hypothesis using two attack methods.
* One of the key claims is that generating an explanation as an intermediate step may improve robustness. The results presented in the table 4 and 5 shows improvement over the baseline.
* The qualitative analysis on the attacks are insightful and provides some intuition for the experiments.

**Summary Of Weaknesses:**

* Surprisingly, the improvement is not consistent across all seq-2-seq methods tested, i.e., some encoder-decoder performs worse than the baseline. Some intuitions on that would be helpful. I would imagine if adding explanations makes the NLI classifier robust, that should generalize to most seq-2-seq setups.
* It is also unclear if the performance's robustness reflects the underlying encoder's robustness. For example, roberta seems to perform better than the other encoders in Table 5. Would the results be similar if the baseline was a roberta classifier instead of BERT?
* Some of the improvements are very low, so it is unclear if that improvement is noise, a variance report would be useful.
* "In order to facilitate (approximate) explanation evaluation, we provide an association between linguistic quality of explanations and model robustness, verified by humans." -- This claim is confusing, and it is unclear what is the evidence to support this claim.

---

### Decision · Program_Chairs · 2024-09-17

**Decision:**

Accept

**Comment:**

Reviewers agree that there is value in the empirical contributions of this paper. The topic is interesting and relevant to the topic of the workshop. That said, Reviewer NrVf gave good feedback that would improve the presentation of the results and the discussion; I recommend taking these into account when preparing the camera-ready.